# The Graph's Apprentice: Teaching an LLM Low-Level Knowledge for Circuit Quality Estimation

## Abstract

Logic synthesis is a crucial phase in the circuit design process, responsible for transforming hardware description language (HDL) designs into optimized netlists. However, traditional logic synthesis methods are computationally intensive, restricting their iterative use in refining chip designs. Recent advancements in large language models (LLMs), particularly those fine-tuned on programming languages, present a promising alternative. This work proposes augmenting LLMs with predictor networks trained to estimate circuit quality directly from HDL code. To enhance performance, the model is regularized using embeddings from graph neural networks (GNNs) trained on Look-Up Table (LUT) graphs, thereby incorporating lower-level circuit insights. The proposed method demonstrates superior performance compared to existing graph-based RTL-level estimation techniques on the established benchmark OpenABCD, while providing instant feedback on HDL code quality.

## 1 Introduction

Rapid technological advancements in computing power has taken an increasingly important role in the past decades in driving scientific research in biology (Schatz, 2012), chemistry (Akimov & Prezhdo, 2015), physics (Dongarra & Keyes, 2024) and especially artificial intelligence, where it has been estimated that at least half of all performance gains in the past ten years have stemmed from hardware improvements alone (Hernandez & Brown, 2020; Dorner, 2021; Karpathy, 2022; Erdil & Besiroglu, 2022; Ho et al., 2024). This ever-rising demand for compute power means that efficient and effective electronic chip design has become increasingly critical.

Modern electronic chip design is a complex, multi-stage endeavor that begins with a chip architect specifying the digital circuit's functionality in a Hardware Description Language (HDL), such as Verilog (Thomas & Moorby, 2008) or VHDL (Coelho, 2012). This HDL code is then subjected to a series of transformations and optimizations, ultimately yielding a physical circuit design that can be manufactured (LaMeres, 2023). In a previous era where circuits were small and limited in functionality, this logic synthesis process was quick and the chip architect could quickly receive feedback and iterate on its HDL code. However, with the increasing complexity of industrial designs, which now can comprise hundreds of millions of logic gates (Amarú et al., 2017), even a single synthesis run has become massively expensive. This has driven the need for alternate ways of providing feedback on HDL code without running the actual logic synthesis process.

A natural way to tackle this problem is to train a machine learning model that can take the HDL code as input, and output estimates of circuit quality such as wire length or delay that could have been computed had the logic synthesis process been run. A few works have approached this topic, by extracting graphical information about the code and using hand-designed statistics of those graphs as features (Zhou et al., 2019; Sengupta et al., 2022; Fang et al., 2023). Although these works had encouraging results, their performance has been limited by the relatively shallow understanding of the semantics of the code that these statistics can provide.

Recently, Large Language Models fine-tuned on code, such as Code-T5 (Wang et al., 2021), Codex (Chen et al., 2021), CodeGen Nijkamp et al. (2023), CodeLlama (Roziere et al., 2023) and DeepSeek-Coder (Guo et al., 2024), have emerged that have proven remarkably successful on a

wide range of tasks (Zheng et al., 2023), most notably as code assistants such as Github Copilot[1]. Although most models are generalists trained on general-purpose programming languages such as C++ and Python, a few models, such as CodeGen-Verilog (Thakur et al., 2023), VeriGen (Thakur et al., 2024), RTLCoder (Liu et al., 2023c) and CodeV (Zhao et al., 2024), have been specifically trained on Verilog, the most popular HDL language. The analysis of these models, however, has been so far limited to investigating their ability to generate realistic code, and an investigation of the predictive power of those internal representations has been lacking.

In this work, we demonstrate for the first time that the hidden states computed by these novel Verilog large language models contain rich insights which can be used to predict quality-of-result metrics with higher accuracy than previous machine learning models. Namely, we feed Verilog code to the state-of-the-art CodeV model, and train an inexpensive decoder neural network that uses the LLM's hidden states as features to predict area and delay. In addition, and critically, we regularize this decoder to encourage its embeddings to resemble those of a graph neural network model trained on Look-Up Table (LUT) graph, an intermediate representation used during the logic synthesis process. The resulting decoder is shown to strongly outperform state-of-the-art baselines, and incidentally shows that those novel Verilog language models extract in their hidden states surprisingly complex insights about the circuits represented by raw code.

Our work makes the following main contributions:

1. We develop the first truly end-to-end machine learning model in the literature, named VeriDistill, which can take raw Verilog code, without any preprocessing, and produce accurate estimates of circuit area/delay metrics.

2. Moreover, we apply during training a novel knowledge distillation method which allows to transfer low-level insights about the circuit, in the form of LUT graphs, back into the machine learning predictor model.

3. We demonstrate through experiments that the combination of those two elements outperforms previous state-of-the-art baselines in a large-scale Verilog dataset and enhances the model's ability to transfer to out-of-distribution data.

4. Finally, we also demonstrate that both using LLM representations and the knowledge distillation are essential, in that removing any one of these components brings the performance back below the previous baselines.

The remainder of this paper is structured as follows. Section 2 provides an overview of the relevant literature and background information. In Section 3, we present a detailed description of our proposed methodology, including its key components and underlying assumptions. The efficacy of our approach is then demonstrated through a series of experiments, which are reported in Section 4. Finally, Section 5 summarizes our main findings, discusses their implications, and outlines potential avenues for future research.

## 2 RELATED WORK

### 2.1 QUALITY-OF-RESULT PREDICTION FROM HDL CODE

Closest to ours is the work of Sengupta et al. (2022). Their approach consists in computing the Abstract Syntax Tree (AST) induced by Verilog code, and extracting from this free vector- and graph-based features. They then train several machine learning models to predict from these features the total negative slack and dynamic power of the circuit. Among all the models evaluated, the XGBoost Regressor performs best and achieves 95% R2-score. The analysis was however limited to different runs of a single circuit and it is not clear how the performance would generalize to different circuits. Since the Abstract Syntax Tree is essentially the raw Verilog code with extra syntactic information, which can be obtained at little cost at inference time by a grammar parser, we include it (along with variants) as baselines in our experimental section.

Further related is the work of Fang et al. (2023) and Fang et al. (2024b). They propose to process Verilog code into a new representation called Simple Operator Graph (SOG), and test several

---

[1]https://github.com/features/copilot

machine learning models (Transformers, Random Forests, Graph Neural Networks and XGBoost regressors) to predict path delay, module-level power and combinatorial area. Although achieving promising results, computing the SOG requires expensive conversion of linguistic data into bit-level operators using logic synthesis tool Yosys (Wolf et al., 2013), which is outside the scope of this work.

Finally, some works take a step further and try to assist circuit design by annotating which parts of HDL is most critical to achieved quality-of-result metrics. For example, Sengupta et al. (2023) attempts to identify timing critical components based on path delay prediction. The AST of each Verilog design is extracted and converted into a graph, with nodes representing IO ports, registers or behavior logic. Behavioral paths are extracted from the graph and used for path-level feature generation. Delay labels of timing paths are generated using commercial synthesis tools, and are assigned to corresponding behavior paths with the same start and end points. By training an XGBoost model on the resulting features, the authors achieve an average classification accuracy of 91%. Also similar is RTL-Timer (Fang et al., 2024a), which ensembles four bit-level circuit representations to predict the post-logic synthesis endpoint arrival time. Such predictions can then be mapped to registers in HDL code to identify critical code paths. Just as in the work of Fang et al. (2023), however, these representations are bit-level rather than word-level and therefore require some degree of processing by logic synthesis tools like Yosys.

## 2.2 LLMS FOR VERILOG

Large language models (LLMs) such as GPT (Ouyang et al., 2022) and Llama (Touvron et al., 2023) have achieved exceptional success in various natural language tasks and have expanded their success to programming languages as well. While decoder-only code LLMs such as Codex (Chen et al., 2021) and CodeLlama (Roziere et al., 2023) have become the most popular due to their exceptional performance in generation tasks like code generation and code translation, older encoder-only models such as CodeBERT (Feng et al., 2020) and encoder-decoder code LLMs such as CodeT5 (Wang et al., 2021) have retained applications in code comprehension tasks such as clone detection and code retrieval.

Although excellent on generalist programming languages like Python or C++, these models have been trained on the relatively small amount of HDL code that is publicly available on the internet, and therefore have performed poorly on Verilog benchmarks like VerilogEval (Liu et al., 2023b) and RTLLM (Lu et al., 2024). This has motivated further work to build LLMs with a higher-degree of knowledge of hardware description languages. Both CodeGen-Verilog (Thakur et al., 2023) and VeriGen (Thakur et al., 2024) used a combination of customized Verilog datasets from code repository website GitHub[2] and various textbooks to fine-tune code LLMs. Finally, RTLCoder (Liu et al., 2023c) used the GPT 3.5 language model (Brown et al., 2020) to generate further Verilog data, in a form of data augmentation, while CodeV (Zhao et al., 2024) used the same model to generate natural language description of real world Verilog code through multi-level summarization.

Besides Verilog code generation from natural language description, LLMs were also explored for other EDA-related tasks. RTLFixer (Tsai et al., 2023) employed Retrieval-Augmented Generation (RAG) and ReAct prompting techniques to interactively debug syntax errors in Verilog code, and achieved remarkable improvement in success rates in the VerilogEval benchmark. ChipNemo (Liu et al., 2023a) explored the application of LLMs in chip design process and adopted several domain adaptation techniques to train an LLM for various applications including assistant chatbots, EDA script generation, and bug summarization and analysis. Finally, ChatEDA (Wu et al., 2024) used code LLMs as an agent to autonomously complete the entire chip design flow from HDL code to the Graphic Data System Version II (GDSII) by managing task planning, script generation and task execution. We refer the reader to the extensive survey of Zhong et al. (2023) for more details on the application of LLMs in electronic design automation and future research directions in this field.

---

[2]www.github.com

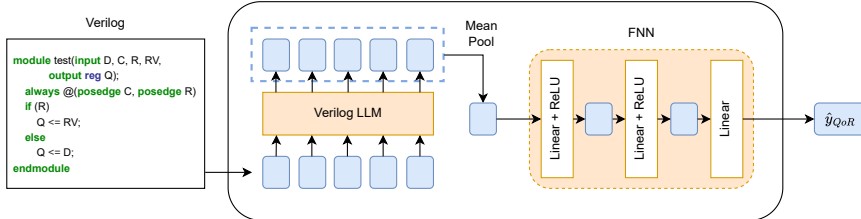

Figure 1: The VeriDistill model. The code input is tokenized and fed to a Verilog-aware Large Language Model (LLM), which produces a sequence of hidden state vectors, one per token. These vectors are averaged, and fed to a small feedforward neural network (FNN) to produce the QoR prediction. In practice, in our experiments we use CodeV-7B (Zhao et al., 2024) as Verilog LLM, and use three layers with ReLU activations in the FNN.

## 2.3 ALIGNMENT OF LLM AND GNN EMBEDDINGS

The multimodal alignment regularizer we propose during training also relates to the broader literature on tuning large language models to align with a pre-trained graph neural network, to incorporate its capabilities.

The work closest to ours is that of Mavromatis et al. (2023), who train a language model to perform a node classification task while adding a regularizer that encourages the predictive distributions to match a pre-trained graph neural network model. The language model makes predictions by passing the graph as input, and extracting the representation corresponding to a final [CLS] classification token. Also similar is Zou et al. (2023), which jointly trains a language model and a graph neural network on a common "context graph prediction" task which encourage alignment of their representations. They then discard the graph neural network and only keep the language model, so that topological characteristics best captured by graph convolutions can be said to have been incorporated in the language model.

More generally, there is a large literature on integrating pretrained graph neural networks with language models by training an adaptive module (Liu et al., 2024; 2023d; Chai et al., 2023; Tang et al., 2024; Cao et al., 2023), allowing the language model to receive inputs from the graph neural network. Alternatively, multiple works have interlaced graph neural network layers and language model layers (Yasunaga et al., 2021; Yang et al., 2021; Zhang et al., 2022; Yasunaga et al., 2022; Jin et al., 2023). In either case, some kind of training is necessary to allow for interactions between the graph neural network and the language model, although the result is not distillation of the graph neural network's perspective into the language model per se.

## 3 METHODOLOGY

We now present our VeriDistill approach in detail. As described in the introduction, turning a high-level description of a circuit in a Hardware Description Language like Verilog into a physical description ready for manufacturing is a computationally expensive process involving several steps, each with an associated intermediate representation describing progressively lower-level elements of the circuit. Our goal is to predict low-level quality-of-result metrics, like area and delay, from the highest-level representation, namely the HDL code.

### 3.1 MODEL

Our model takes as input Verilog code, which is fed to a Large Language Model (LLM). This LLM has been specifically fine-tuned on Verilog code generation. The code is first split into a sequence of tokens, which are then fed in parallel in the LLM. As an output, the LLM produces a sequence of high-dimensional "hidden state" vectors, one for each token that is inputted to the LLM. We average these hidden states, producing a single vector. This vector is then fed to a feedforward neural network, composed of several linear layers with nonlinear activations, which finally outputs the QoR estimate. A diagram is provided as Figure 1.

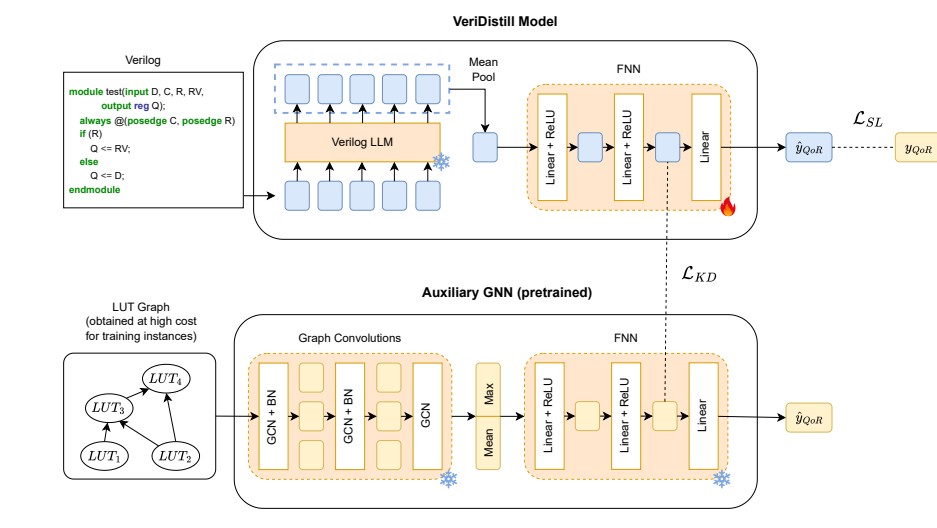

Figure 2: The training procedure. The Verilog training examples are passed to the VeriDistill model, which produces predictions of the QoR metric. These predictions are scored against the true QoR values by a mean-squared error supervised learning loss. In addition, the LUT graph representation resulting from logic optimization is fed to an auxiliary GNN model, pretrained to perform the same QoR prediction task. The hidden representations at the last layer of both the VeriDistill and GNN models is extracted, and a mean-square error knowledge distillation loss encourages these two representations to be similar, despite having different inputs. Both the pretrained GNN and LLMs modules are kept frozen during training.

## 3.2 TRAINING

We train the model as follows. We assume we have access to a training set of circuits with Verilog code for which the expensive logic synthesis process has been performed, so that we know their QoR metric (such as area or delay). In addition, as an intermediate product of the logic synthesis process, an LUT graph is produced immediately following the logic optimization phase, which we save. This yields a collection of training triples $\mathcal{D} = \{(X_{\text{Verilog}}, X_{\text{LUT}}, y_{\text{QoR}})\}$.

### 3.2.1 SUPERVISED LEARNING

Given such a dataset, we treat our problem by supervised machine learning. The LLM, which has been pretrained on Verilog code, is kept frozen, so that only the FNN gets updated. In a training step, the Verilog code $X_{\text{Verilog}}$ is fed to the VeriDistill model to produce a prediction $\hat{y}_{\text{QoR}}$. This prediction is compared in mean-squared error loss with the true QoR metric $y_{\text{QoR}}$ as a supervised learning loss

$$\mathcal{L}_{\text{SL}} = \left(\hat{y}_{\text{QoR}} - y_{\text{QoR}}\right)^2. \tag{1}$$

### 3.2.2 LOW-LEVEL KNOWLEDGE DISTILLATION

In practice, training only with the supervised learning loss leads to limited performance. One potential explanation is that there is too much of a gap between a high-level circuit description like Verilog and the low-level metrics we purport to predict. Intuitively, to perform high-quality predictions, we would want the model to possess some degree of understanding of lower-level circuit design while still only taking Verilog code as input.

We propose the following approach to address this problem. Prior to training, we pretrain a Graph Neural Network (GNN) to predict the same QoR metric as VeriDistill, but from the Look-Up-Table (LUT) graph $X_{\text{LUT}}$ of the circuit obtained after optimization using Yosys (Wolf et al., 2013). This graph, which can be seen as an alternative to the more popular And-Inverter Graph (AIG) format, is particularly suitable for GNN training as it is compact with rich node information. Moreover, as a circuit representation, it sits intermediate between a high-level description of the circuit encoded

in the Verilog code, and a physical circuit description from which the QoR metrics such as area and delay can be read. Prediction from LUT graphs is thus easier than prediction from Verilog code, but not completely trivial either.

The GNN architecture we adopt is composed of a sequence of graph convolutions, followed by joint mean and max pooling, and a sequence of linear layers. We pretrain it using the supervised learning loss (1) until good predictive performance is achieved. Then, during the VeriDistill training, we keep the GNN weights frozen and we propose to encourage the last-layer activations of the VeriDistill model $z_{\text{VeriDistill}}^{(-1)}$ to resemble those of the GNN model $z_{\text{GNN}}^{(-1)}$, despite these models operating on different inputs. We perform this simply by adding a mean-square error loss

$$\mathcal{L}_{\text{KD}} = \left\| z_{\text{VeriDistill}}^{(-1)} - z_{\text{GNN}}^{(-1)} \right\|_2^2 \tag{2}$$

in the total loss. As the weights of the GNN are pretrained and kept frozen while the VeriDistill model is being trained, this is effectively a form of knowledge distillation from the GNN to the VeriDistill model.

### 3.2.3 TOTAL LOSS

We balance the importance given to the knowledge distillation compared to the supervised learning objective using a hyperparameter factor $\alpha$, yielding the final loss

$$\mathcal{L} = \alpha \mathcal{L}_{\text{SL}} + (1 - \alpha) \mathcal{L}_{\text{KD}}.$$

A diagram describing the VeriDistill training process is provided as Figure 2.

## 4 EXPERIMENTS

This section is organized as follows: We begin by presenting the implementation details of our experimental setup in Section 4.1, including hardware, model, and training hyperparameters. Next, we describe the dataset used and the data preprocessing steps for training and evaluation in Section 4.2. We then introduce the baseline methods and their implementation details in Section 4.3. Finally, we present the results on the main datasets and a study on unseen out-of-distribution circuits in Sections 4.4 and 4.5.

### 4.1 EXPERIMENTAL SETUP

For our experiments, we use the following implementation of the model. We use CodeV-7B (Zhao et al., 2024) as Verilog LLM, and use three layers with ReLU activations in the feedforward neural network. The model takes as input strings, which are broken into a sequence of the 32,016 possible tokens in CodeV-7B's vocabulary. The language model processes these inputs into a sequence of the same length, made up of 512-dimensional vectors. After mean pooling, the resulting vector is passed to the feedforward neural network, which uses 512-dimensional activations, before making the final prediction. In particular, this architecture means that the last-layer activations $z_{\text{VeriDistill}}^{(-1)}$ are 512-dimensional.

The auxiliary GNN teacher model takes a LUT graph with 16-dimensional node attributes, and passes it through three 64-dimensional graph convolutional layers interleaved with batch normalization layers. After concatenation of the mean and max pooling outputs, the 128-dimensional vector is passed through three 512-dimensional linear layers with ReLU activations before the final prediction. Thus, in particular, the last-layer activations $z_{\text{GNN}}^{(-1)}$ are 512-dimensional, matching with those of the VeriDistill model.

We implement VeriDistill and the baselines using the `PyTorch` and `PyG` libraries. Models which do not use our knowledge distillation procedure are trained using the ReduceLROnPlateau scheduler with initial learning rate 1e-3, patience set to 30 epochs and factor set to 0.5. In contrast, models involving our knowledge distillation procedure are trained using the CosineAnnealingLR Loshchilov & Hutter (2017) scheduler, with an initial learning rate of 1e-3 and number of iterations set to 50. We start the training process with $\alpha = 0.5$, and increase $\alpha$ to 0.75 and 1 at epochs 150 and 250. The idea is put less emphasis on knowledge distillation at every warm re-start. We find that this approach

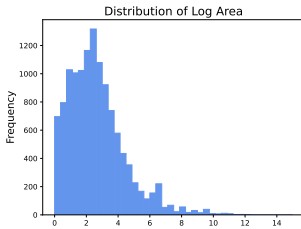 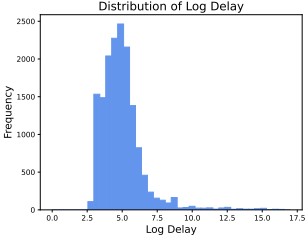 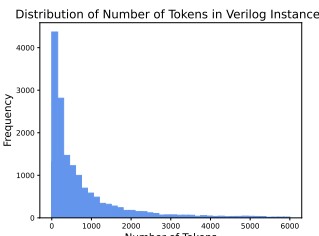

Figure 3: Distribution of labels and the number of tokens in the Verilog dataset.

results in marginal gain compared to other optimization methods. All models are trained until full convergence. Details about the training resources and times can be found in Appendix C.

OpenROAD provides two optimization recipes for the logic synthesis process: "ABC_AREA=1" for area optimization and "ABC_SPEED=1" for timing optimization. The results reported under Section 4.4 are produced under area optimization. We report the results under delay optimization in Appendix D. we find that our approach works as well under different recipe optimization settings.

## 4.2 DATASETS

We train and evaluate on two separate datasets. The first dataset is used for training, validation, and testing of all the methods, while OpenABCD contains out-of-distribution circuits aiming to challenge VeriDistill and determine its ability to generalize.

**Customized Dataset**   To train and evaluate our proposed method, we collect 18.4k Verilog examples provided by Pei et al. (2024) and 5.8k from Thakur et al. (2022). These Verilog examples are obtained from open-source GitHub repositories and textbooks and have been verified for syntax correctness. We use an open-sourced EDA platform OpenROAD Ajayi et al. (2019) with 7nm technology PDK provided to conduct logic synthesis and record post-synthesis labels of area and delay. We convert the AIG graphs obtained after logic optimization into LUT graphs and save them for training the auxiliary GNN model.

Note that a substantial fraction of the code snippets end up being functionally incorrect and failing some stage of the logic synthesis pipeline. Since we require functionally correct examples for their QoR metric to be well-defined, we removed such examples during the preprocessing. In addition, although not strictly a problem for our method, one of the competing baselines requires extracting the Abstract Syntax Tree (AST) of the Verilog, which is obtained by running a parser on the code. The parser was unable to produce AST representations for a small fraction of the instances (FRACTION%), which we removed from consideration. The resulting dataset, after filtering bad examples, ended up having 16k examples, which we split into training, validation, and test sets with a ratio of 0.75/0.1/0.15, respectively.

We depict the distribution of labels and the number of tokens in Verilog instances in Figure 3. As mentioned in prior work by Zhao et al. (2024) , Verilog data scarcity is a common challenge in developing machine learning tools for RTL level tasks. We note that the majority of Verilog instances contain less than 2000 tokens, with the corresponding circuits having a small area and delay.

**OpenABCD**   Additionally, we consider data provided by Chowdhury et al. (2021) to evaluate the transferability of our method to unseen circuits. The OpenABCD dataset consists of functionally diverse designs such as bus communication protocols, computing processors, digital signal processing cores, cryptographic accelerators and system controllers.

## 4.3 BASELINES

While numerous prior works have attempted to predict post-synthesis circuit quality at the RTL-stage, none of them perform prediction directly from source Verilog files. Several works rely on

| Method | Area | | | | Delay | | | |
|---|---|---|---|---|---|---|---|---|
| | MAE ↓ | R2 ↑ | MAPE ↓ | RSE ↓ | MAE ↓ | R2 ↑ | MAPE ↓ | RSE ↓ |
| LUT-GNN (Teacher) | 0.280 | 0.933 | 0.437 | 0.067 | 0.251 | 0.918 | 0.050 | 0.082 |
| AST-XGBoost | 0.773 | 0.745 | 1.494 | 0.362 | 0.521 | 0.632 | 0.096 | 0.565 |
| AST-GNN | 0.867 | 0.660 | 1.365 | 0.34 | 0.622 | 0.520 | 0.116 | 0.480 |
| AST-GNN w/ KD | 0.898 | 0.670 | 1.327 | 0.33 | 0.654 | 0.561 | 0.122 | 0.439 |
| CodeV + Decoder | 0.991 | 0.614 | 1.901 | 0.386 | 0.718 | 0.443 | 0.141 | 0.557 |
| VeriDistill | **0.495** | **0.862** | **0.629** | **0.138** | **0.415** | **0.728** | **0.076** | **0.272** |

Table 1: The performance of different Verilog models on the test dataset, where the best result for each metric is bolded. In addition, we report the performance of the teacher model trained on the LUT graphs, which serves as an upper-bound.

lower-level circuit representation that requires extra processing using logic synthesis tools (Zhou et al., 2019; Fang et al., 2023). Using low-level circuit representation as input is advantageous for the circuit quality prediction task but it is unfair to compare them to our method which takes unprocessed Verilog as input, as reliance on external processing tools makes their computation fragile and in some cases prohibitively expensive.

We adopt the method proposed by Sengupta et al. (2022) as our baseline. It relies on AST representations that can be easily converted from Verilog source files. We implement the method based on description in Sengupta et al. (2022). Verilator (Snyder, 2004) is used to convert each source Verilog into its respective AST representation, which can be represented as a graph. The nodes in the graph represent one of the following five semantic categories from the source Verilog (`root`, `variable`, `operation`, `constant`, `edge`), while edges are created between nodes with connections.

We implement three variants of the AST-based method:

**AST-XGBoost** We compute the following features: $(i)$ the total number of input bits, $(ii)$ the total number of output bits, $(iii)$ the longest path in the AST, $(iv)$ the frequency of each node type in the graph and $(v)$ the frequency of each logic type in the graph. The features are concatenated to form a feature vector with 108 features [3]. We perform a thorough hyper-parameter selection using grid search and employ early stopping to prevent over-fitting.

**AST-GNN w/o KD** The AST-GNN model takes in the following features per node: $(i)$ the total number of input bits, $(ii)$ the total number of output bits, $(iii)$ the node semantic type and $(iv)$ the node operation type. Each feature is represented via a one-hot vector and is projected to a 4-dimensional space via a linear layer. The final node features consist of a $(4 \times 4) = 16$-dimensional vector. We cap the number of input/output bits to 200, since 99.9 percent of the nodes in the dataset have less than 200 input/outputs. The AST-GNN model utilizes the same hyperparameters and architecture as the auxiliary GNN model used for the knowledge distillation objective in VeriDistill.

**AST-GNN w/ KD** We propose a third baseline, where the AST-GNN model is guided by the LUT GNN model. The baseline utilizes the same student-teacher knowledge distillation as our method. We introduce this baseline to demonstrate the effectiveness of utilizing an LLM in the student network.

### 4.4 MAIN RESULTS

We first summarize the results of our main experiment, where we train and test the model on the large Customized Dataset (see Section 4.2). Table 1 outlines the performance of different models on the test set. As can be seen, our proposed method, utilizing both CodeV as an encoder and knowledge distillation, outperforms other baselines across all the metrics, especially with area prediction. Interestingly, simply using a decoder on the LLM representation performs worse than the

---

[3] $108 = 1 + 1 + 1 + 5 + 100$ features coming from feature categories $(i)....,(v)$

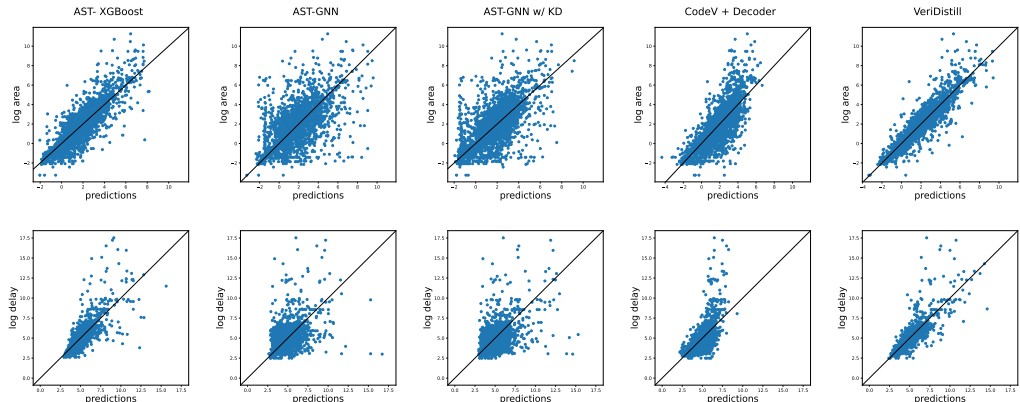

Figure 4: Prediction vs. target on test data. The predicted values using different methods are plotted against the targets. (Top) Area prediction. (Bottom) Delay prediction.

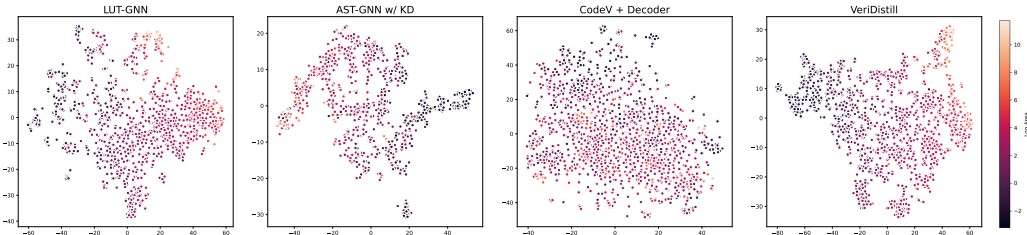

Figure 5: t-SNE representation of the last hidden representation of models on the test data. Color represents the target value (log-area).

previous state-of-the-art, while knowledge distillation on the AST-GNN model has almost no effect. Only when both are used together is there profound impact on performance, which suggests our knowledge distillation procedure is crucial in fully exploiting the richness of the CodeV LLM representations.

We can further insight on the benefits of our combined approach by analyzing scatter plots of the predictions against the targets, shown in Figure 4. As can be seen, most models' good performance is mostly concentrated on circuits with small delay and area, at the expense of larger circuits, perhaps because the latter are more rare in the training set. In contrast, our model performs mostly uniformly well on circuits on every size. This contrast is particularly pronounced when comparing against the same model without knowledge distillation (CodeV+Decoder), which indicates that our knowledge distillation procedure is crucial in allowing our model to perform well across the whole range of circuit sizes.

Finally, in Figure 5, we present the t-SNE projection of the last hidden space representations on the test data from the teacher model ($Z_{teacher}$) trained for predicting log-area, alongside those from the LLM-based models. As can be seen, the resulting t-SNE representation of the VeriDistill model appears very similar to the one of the LUT-GNN teacher model. Most importantly, the t-SNE of the LUT-GNN model appears to have captured a clear left-to-right pattern in log-area, which shows that the teacher model's representations have captured a very precise prediction pattern for log-area. This linear pattern has been transferred just as well to VeriDistill. On the contrary, the t-SNE projection of the AST-GNN w/ KD does not exhibit the same vivid pattern as VeriDistill does, where the homogeneity of clusters is abrupter by points of different colors. Finally, the plot of the CodeV + Decoder appears much more like an undefined mass, where the log-area values are mixed together indiscriminately.

| IP | IO | Nodes | Edges | Area (MAE ↓) | | Delay (MAE ↓) | |
|---|---|---|---|---|---|---|---|
| | | | | w/o KD | w/ KD | w/o KD | w/ KD |
| spi | 492 | 4219 | 8676 | 2.083 | **0.893** | **0.049** | 0.053 |
| sasc | 260 | 613 | 1351 | **0.738** | 1.375 | 0.284 | **0.319** |
| i2c | 305 | 1169 | 2466 | 1.986 | **1.662** | **0.329** | 0.571 |
| simple_spi | 296 | 930 | 1992 | **0.816** | 1.142 | 0.553 | **0.071** |
| wb_conmax | 4197 | 47840 | 97755 | 4.807 | **3.541** | **0.372** | 1.312 |
| vga_lcd | 34385 | 105334 | 227731 | 6.109 | **5.063** | 0.238 | **0.021** |
| aes_secworks | 5691 | 40778 | 84160 | 4.043 | **3.434** | 0.604 | **0.33** |
| sha256 | 2985 | 15816 | 32647 | 2.374 | **1.749** | **0.4** | 1.46 |
| ss_pcm | 194 | 462 | 896 | 0.844 | **0.367** | **0.413** | 0.462 |
| fir | 761 | 4558 | 9467 | 2.455 | **1.132** | 0.718 | **0.25** |
| idft | 75022 | 241552 | 520523 | 5.975 | **4.494** | 0.379 | **0.258** |
| des3_area | 367 | 4971 | 10006 | 2.828 | **1.298** | **0.441** | 0.467 |
| ethernet | 21153 | 67164 | 144750 | 6.32 | **5.743** | **0.52** | 0.62 |
| dft | 75014 | 245046 | 527509 | 5.999 | **4.576** | 0.347 | **0.152** |
| dynamic_node | 5283 | 18094 | 38763 | 5.793 | **5.616** | 1.307 | **1.244** |
| tv80 | 997 | 11328 | 23017 | 5.049 | **2.544** | 1.544 | 0.864 |
| pci | 6586 | 19547 | 42251 | 4.392 | **2.303** | **0.163** | 0.668 |
| fpu | 1041 | 29623 | 59655 | 4.326 | **2.519** | 2.301 | **1.275** |
| usb_phy | 222 | 487 | 1064 | 1.682 | **1.266** | 0.283 | **0.007** |
| aes_xcrypt | 3780 | 45840 | 93485 | 5.493 | **3.786** | 1.506 | **0.795** |
| iir | 935 | 6978 | 14397 | 2.585 | **2.026** | 0.493 | **0.301** |
| aes | 1212 | 28925 | 58379 | 3.912 | **1.43** | 0.321 | **0.181** |
| mem_ctrl | 2149 | 16307 | 37146 | 3.504 | **2.397** | 0.609 | **0.28** |
| Avg. | | | | 3.657 | **2.624** | 0.616 | **0.520** |

Table 2: OpenABCD results. VeriDistill with or without KD have been trained on customized datasets and used to predict post-synthesis area and delay of OpenABCD circuits without any fine-tuning. Mean Absolute Error (MAE) between estimated and actual logarithmic values are reported for area and delay. IO, Node and Edges represent the number of primary inputs/outputs, AIG nodes and AIG edges of the circuits.

### 4.5 ADDITIONAL OUT-OF-DISTRIBUTION RESULTS

Finally, we evaluate how our knowledge-distillation procedure can impact the ability of the trained model to generalize to new out-of-distribution circuits. For this, we take our model, trained with and without knowledge distillation on our Customized Dataset, and apply it to instances in the Open-ABCD benchmark (see Section 4.2). As can be seen in Table 2, our knowledge distillation procedure systematically improves the LLM-based model's ability to transfer prediction performance on out-of-distribution instances, which differ significantly from those seen during training.

## 5 CONCLUSION

In summary, in this work we propose a novel procedure to predict quality-of-result electronic circuit metrics from Verilog code, by training a small neural network model on Verilog LLM representations with a knowledge distillation regularizer which align its internal activations with those of a low-level GNN model. We show that this new model, which we call VeriDistill, outperforms previous approaches in prediction accuracy.

Besides the clear practical value of our method, our results highlight the surprising phenomenon that Verilog LLMs appeared to have learned more abstract characteristics regarding the circuit represented by the code, which can be exploited to predict ultimate circuit quality with higher accuracy than any previous method. In essence, Verilog LLMs might have learned to do a mini "logic synthesis", despite having only been trained to perform language modeling. However, our results also highlight the importance of our knowledge distillation procedure in allowing downstream models to effectively use this information stored in the LLM's representations.

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

# 6 APPENDIX

APPENDIX A: DIFFERENT LARGE LANGUAGE MODELS

To demonstrate performance of VeriDistill with different LLMs, we employ CodeV-DeepSeek and CodeV-CodeQwen, which utilize `deepseek-coder-6.7b` and `CodeQwen1.5-7B-Chat` as the base models. These two models are two variants of the standard CodeV model based on `CodeLlama-7b-Instruct`, and were trained with the same procedure.

| Approach | Area | | | | Delay | | | |
|---|---|---|---|---|---|---|---|---|
| | MAE ↓ | R2 ↑ | MAPE ↓ | RSE ↓ | MAE ↓ | R2 ↑ | MAPE ↓ | RSE ↓ |
| CodeQwen + Decoder | 1.070 | 0.563 | 1.975 | 0.437 | 0.732 | 0.368 | 0.139 | 0.632 |
| DeepSeek + Decoder | 1.061 | 0.566 | 2.184 | 0.434 | 0.738 | 0.367 | 0.143 | 0.633 |
| CodeV + Decoder | 0.991 | 0.614 | 1.901 | 0.386 | 0.718 | 0.443 | 0.141 | 0.557 |
| VeriDistill (CodeQwen) | 0.468 | 0.878 | 0.574 | 0.122 | 0.424 | 0.733 | 0.078 | 0.267 |
| VeriDistill (DeepSeek) | 0.484 | 0.875 | 0.622 | 0.125 | 0.426 | 0.706 | 0.077 | 0.294 |
| VeriDistill (CodeV) | 0.495 | 0.862 | 0.629 | 0.138 | 0.415 | 0.728 | 0.076 | 0.272 |

Table 3: The performance of VeriDistill with different Large Language Models.

APPENDIX B: ADDITIONAL RESULTS ON THE OPENABCD BENCHMARK

| IP | Area (MAE ↓) | Delay (MAE ↓) |
|---|---|---|
| spi | 0.294 | 1.218 |
| sasc | 0.035 | 0.708 |
| i2c | 0.21 | 0.867 |
| simple_spi | 0.613 | 0.637 |
| wb_conmax | 2.343 | 1.104 |
| vga_lcd | 0.104 | 6.037 |
| aes_secworks | 2.671 | 1.129 |
| sha256 | 0.887 | 1.399 |
| ss_pcm | 0.581 | 1.229 |
| fir | 0.391 | 0.325 |
| idft | 1.018 | 7.471 |
| des3_area | 0.287 | 1.797 |
| ethernet | 0.097 | 4.567 |
| dft | 1.032 | 7.258 |
| dynamic_node | 1.131 | 0.966 |
| tv80 | 0.05 | 1.399 |
| pci | 0.11 | 2.978 |
| fpu | 0.155 | 0.303 |
| usb_phy | 0.115 | 0.85 |
| aes_xcrypt | 2.227 | 1.372 |
| iir | 0.353 | 0.456 |
| aes | 0.917 | 3.463 |
| mem_ctrl | 0.922 | 1.327 |
| Avg. | 0.103 | 0.105 |

Table 4: The performance on LUT-GNN (teacher model) on the OpenABCD benchmark.

APPENDIX C: TRAINING RESOURCES

Since the LLM is kept frozen during training, it was possible to save training time by extracting the forward pass through the LLM only once and saving it. We performed this phase on a machine with 8 Nvidia V100 GPUs with 32GB of memory and 32 Intel(R) Xeon(R) Gold 6140 CPUs. Once the

| Method | Training Time (Till Convergence) | Number of Epochs to Converge |
|---|---|---|
| LUT-GNN | 21 hours | 300 |
| AST-XGBoost | 5 minutes | N/A |
| AST-GNN | 33 minutes | 340 |
| AST-GNN w/ KD | 40 minutes | 300 |
| CodeV + Decoder | 12 hours | 360 |
| VeriDistill | 18 hours | 260 |

Table 5: Training times for the various models.

hidden state a then trained each model following the procedure detailed in the paper on the same machine using a single V100 GPU with 1024 minibatch sizes. The training times for each model are summarized in the following table.

APPENDIX D: RESULTS UNDER DIFFERENT SYNTHESIS SETTING

To test the robustness of VeriDistill under a different synthesis setting, we re-run synthesis for speed optimization (ABC_SPEED=1 for OpenROAD hyperparameter setting). We train and evaluate all the methods under the new setup.

| method | Area | | | | Delay | | | |
|---|---|---|---|---|---|---|---|---|
| | MAE ↓ | R2 ↑ | MAPE ↓ | RSE ↓ | MAE ↓ | R2 ↑ | MAPE ↓ | RSE ↓ |
| LUT-GNN (Teacher) | 0.251 | 0.955 | 0.309 | 0.045 | 0.109 | 0.948 | 0.023 | 0.052 |
| AST-XGBoost | 0.749 | 0.745 | 1.366 | 0.349 | 0.484 | 0.652 | 0.093 | 0.542 |
| AST-GNN | 0.893 | 0.661 | 1.435 | 0.339 | 0.317 | 0.604 | 0.071 | 0.396 |
| AST-GNN w/ KD | 0.872 | 0.674 | 1.418 | 0.331 | 0.324 | 0.621 | 0.082 | 0.392 |
| CodeV + Decoder | 0.991 | 0.629 | 1.69 | 0.371 | 0.367 | 0.533 | 0.086 | 0.467 |
| VeriDistill | 0.482 | 0.872 | 0.784 | 0.128 | 0.236 | 0.781 | 0.054 | 0.219 |

Table 6: The performance of different Verilog models on the test dataset under the speed optimization setting.

APPENDIX E: DISTRIBUTION OF ABSOLUTE PERCENTAGE ERRORS

To compliment the results in Figure 4, Tables 7 and 8 outline the distribution of the absolute percentage errors of each method. Each cell specifies the number of points with the absolute percentage error falling in the range specified by the column.

| Method | 0 - 0.1 | 0.1 - 0.3 | 0.3 - 0.5 | 0.5 - 1.0 |
|---|---|---|---|---|
| AST-XGBoost | 425 | 654 | 334 | 452 |
| AST-GNN | 516 | 496 | 305 | 548 |
| AST-GNN w/ KD | 355 | 583 | 369 | 558 |
| CodeV + Decoder | 281 | 558 | 427 | 599 |
| VeriDistill | 781 | 585 | 217 | 282 |

Table 7: Distribution of absolute percentage errors ($\frac{|prediction-label|}{label}$) for the (log) area prediction task.

| Method | 0 - 0.1 | 0.1 - 0.3 | 0.3 - 0.5 | 0.5 - 1.0 |
|---|---|---|---|---|
| AST-XGBoost | 1572 | 695 | 81 | 26 |
| AST-GNN | 1402 | 799 | 141 | 33 |
| AST-GNN w/ KD | 1298 | 891 | 153 | 30 |
| CodeV + Decoder | 1127 | 1002 | 192 | 47 |
| VeriDistill | 1803 | 504 | 52 | 17 |

Table 8: Distribution of absolute percentage errors ($\frac{|prediction-label|}{label}$) for the (log) delay prediction task.

