# OpenReview forum: "The Graph's Apprentice: Teaching an LLM Low-Level Knowledge for Circuit Quality Estimation"
_ICLR.cc/2025/Conference — Submitted to ICLR 2025_

### Official Review · Reviewer_zReM · 2024-10-26

**Soundness:** 3
**Presentation:** 4
**Contribution:** 3
**Rating:** 5
**Confidence:** 5

**Summary:**

This paper proposes a framework, which utilize a GNN trained on LUT netlist-level graph as a teacher in order to transfer knowledge to LLM encoder based circuit quality estimator. It enables model to get more accurate performance and area of circuits end-to-end with RTL as input.

**Strengths:**

This paper is the first to implement knowledge transfer of circuit quality estimation from netlist-level to RTL-level. Considering that netlist-level circuit performance and area estimation is more precise and RTL-level circuit quality prediction is faster, this work strike a balance between the precision and inference speed.

**Weaknesses:**

In this work, the LLM is used as an encoder, leveraging its embeddings to predict circuit quality. However, a potential limitation arises from the token count restrictions of LLMs, which may prevent the framework from handling large Verilog designs. For smaller designs, which can be processed within these token constraints, transforming them into netlists is relatively quick and incurs minimal computational cost. Given this, why not employ a GNN to process the netlist representation directly? Predictions at the netlist level are generally more accurate than those at the RTL level, making this a potentially more effective approach. Moreover, the authors mainly compare there work with models trained on AST. Such an experiment lacks persuasive power.
Things to improve: The ‘Fig. ??’ in section 4.1.

**Questions:**

1.Given the token count limitations of LLMs, which may prevent the framework from handling large Verilog designs, and the relatively quick and low-cost process of transforming smaller designs into netlists, wouldn’t it be more effective to use a GNN to process the netlist representation directly?
2.Why transform aiger into LUT? What’s the advantage of LUT? And in Advanced integrated circuit (IC) manufacturing process, whether LUT is widely used?

---

> ### Author Response · Authors · 2024-11-26
>
> We will answer together the concerns of the weakness section, and those of question 1, as they are closely linked.
>
> Small vs large designs and Question 1: This is an excellent point. To clarify our goals, the use-case we have in mind is precisely large circuits for which computing the netlist is too slow to be used to provide near-immediate feedback to a circuit designer. It is also why we restricted ourselves to comparisons with ASTs, since they are some of the few alternatives to raw Verilog that can be obtained fast enough to fit this use-case. We hope this clears any misunderstandings.
>
> Regarding LLM context length, we used the Verilog LLM with the largest context length that is publicly available (16k tokens). Most importantly, in a real practical application, we would expect that a company interested in implementing the method could fine-tune a larger model if desired. Indeed, the latest LLMs now boast context lengths so large that context length does not become a constraint anymore: as extreme examples, the huggingface model repository now stores an increasing number of large context open-source LLMs that can handle above 1M tokens (e.g. Llama-3-8B-Instruct-Gradient-4194k by gradient.ai), which at 30 tokens/line of code would represent over 33k lines of code. Although such models has not been fine-tuned on Verilog code, there would not be any obstacle in principle for an interested company to do so, now that there is a use-case for it.
>
> Question 2: This is an excellent question. In principal, either AIG or LUT could be used to train the teacher model. In our initial experiments conducted on a subset of simpler circuits, we found that the GNN trained on LUT outperforms the GNN trained on AIG by a small margin. We suspect this was the case since graph neural networks perform best when graphs are small and node features are rich, and LUT graphs fit closer to this ideal than AIGs. In addition, the LUT representation of the circuit has more resemblance to the final Netlist. From a practical perspective, the LUT is more compact in representation , with up to 4x less nodes than the AIG. Thus, it is easier to fit large circuits into the memory with the LUT representation.
>
> Proofreading: We went over the paper thoroughly and fixed every typo we could find. Thanks for pointing it out.

---

> ### Author Response · Authors · 2024-12-02
>
> As the discussion period nears its end, we hope that we have effectively addressed and resolved your concerns. If so, we would be grateful if you could reconsider your rating of our paper.

---

### Official Review · Reviewer_fovb · 2024-10-29

**Soundness:** 3
**Presentation:** 3
**Contribution:** 3
**Rating:** 6
**Confidence:** 4

**Summary:**

This paper presents VeriDistill, the first end-to-end machine learning model that processes raw Verilog code to predict circuit quality metrics. The model uses an innovative knowledge distillation technique, transferring low-level circuit insights via LUT graphs into an LLM-based predictor. Experiments show that VeriDistill surpasses current state-of-the-art methods on large-scale Verilog datasets and demonstrates transferability to out-of-distribution datasets.

**Strengths:**

This work introduces VeriDistill, an innovative end-to-end machine learning model that predicts circuit quality directly from raw Verilog code, which is an important and challenging task in the domain of electronic design automation (EDA). It applies a novel knowledge distillation method, transferring low-level circuit insights via LUT graphs into an LLM-based predictor. And VeriDistill demonstrates robust performance, outperforming state-of-the-art baselines on large-scale and out-of-distribution datasets.

**Weaknesses:**

There are several limitations for VeriDistill. Firstly, as shown in Table 1, VeriDistill’s performance lags significantly behind the teacher model. To improve this, I think a possible way is to unfreeze some parameters in the base LLM and fine-tune them with the decoder. Secondly, the paper only uses CodeV as its base model without exploring other Verilog LLMs that might provide different insights. Conducting more experiments with various models could better demonstrate the method's effectiveness and robustness. Additionally, the paper needs careful proofreading; for instance, there is a missing figure referenced in Line 293.

**Questions:**

1. For Table 1, you state that knowledge distillation has almost no effect on the AST-GNN model. But in Figure 5, you show that the knowledge distillation makes the t-SNE representation of AST-GNN more like that of the teacher model. These statements seem contradictory and confusing. Could you provide further clarification?
2. For Table 2, could you also report the performance of the teacher model on the out-of-distribution dataset? This would help clarify the current gap in performance.
3. It seems VeriDistill can only handle small Verilog designs due to the context window length constraint of the LLM base model. Is there a way to scale VeriDistill for larger designs? Given that logic synthesis for small designs is usually fast and the error remains significant, the current model may not be very attractive.
4. When the logic synthesis recipe changes, the delay and area metrics of circuits typically change as well. How can VeriDistill handle these variations?

---

> ### Author Response · Authors · 2024-11-26
>
> Thank you very much for your review. We will address each point in turn:
>
> Layer unfreezing: This is an excellent suggestion! Unfortunately, we do not currently possess the computational resources to undertake this experiment, but it could be an interesting way to potentially push the performance of our approach even further.
>
> Different LLMs: Absolutely, we agree this could be a concern! To demonstrate the impact of KD with respect to the base LLM, we employ CodeV-DeepSeek and CodeV-CodeQwen, which utilize deepseek-coder-6.7b and CodeQwen1.5-7B-Chat as the base models. These two models are two variants of CodeV, which have undergone the same Verilog fine-tuning procedure as CodeV-codeLlama. These two models are two variants of the standard CodeV model based on CodeLlama-7b-Instruct, and were trained with the same procedure. As can be seen in the table in Appendix A, the resulting performance only varied marginally after switching to those different LLMs.
>
> Proofreading: We made a new pass and corrected everything.
>
> Question 1: We would like to clarify that by stating "knowledge distillation has almost no effect on the AST-GNN model", we are referring to the performance of the model. As shown in the results, the performance of AST-GNN + KD does not improve significantly over AST-GNN. Furthermore, the tSNE figure is a bit misleading due the fact that the latent representation space is projected to 2D from 512D. Still, the t-SNE shows that while AST-GNN with KD aligns with the teacher model, there are some mixups in the points visible. We have modified the text to make this point more clear.
>
> Question 2: We have added this result to Appendix B.
>
> Question 3: It is true that the current choice of model takes a context of 16k tokens, which corresponds to about 500 lines of Verilog code. However, there has been a steady stream of new models coming out with increasingly larger contexts - for example, the huggingface model repository now stores an increasing number of large context open-source LLMs that can handle above 1M tokens (e.g. Llama-3-8B-Instruct-Gradient-4194k by gradient.ai). Thus, although the current Verilog LLMs are smaller context, there is no obstacle to fine-tuning, in principle, larger LLMs to fit the use case would the need be.
>
> Question 4:  Currently, the model is effectively trained to produce QoR estimates conditional on a fixed recipe, because of the way the labels were produced during data generation (i.e. always using the same fixed recipe). OpenROAD provides two optimization recipes for the logic synthesis process: "ABC\_AREA=1" for area optimization and "ABC\_SPEED=1" for timing optimization. The experiments in the paper adopted area optimization recipe. To test the robustness of VeriDistill under a different synthesis setting, we re-run synthesis for speed optimization (ABC\_SPEED=1 for OpenROAD hyperparameter setting).  We train and evaluate all the methods under the new setup. As seen in the table below, all our claims in the paper stand well. We have added this table to Appendix D.
>
> The performance of different Verilog models on the test dataset under the speed optimization setting:
> | Method               | MAE ↓   | R2 ↑    | MAPE ↓  | RSE ↓   | MAE ↓   | R2 ↑    | MAPE ↓  | RSE ↓   |
> |----------------------|---------|---------|---------|---------|---------|---------|---------|---------|
> | LUT-GNN (Teacher)    | 0.251   | 0.955   | 0.309   | 0.045   | 0.109   | 0.948   | 0.023   | 0.052   |
> | AST-XGBoost          | 0.749   | 0.745   | 1.366   | 0.349   | 0.484   | 0.652   | 0.093   | 0.542   |
> | AST-GNN              | 0.893   | 0.661   | 1.435   | 0.339   | 0.317   | 0.604   | 0.071   | 0.396   |
> | AST-GNN w/ KD        | 0.872   | 0.674   | 1.418   | 0.331   | 0.324   | 0.621   | 0.082   | 0.392   |
> | CodeV + Decoder      | 0.991   | 0.629   | 1.69    | 0.371   | 0.367   | 0.533   | 0.086   | 0.467   |
> | VeriDistill          | 0.482   | 0.872   | 0.784   | 0.128   | 0.236   | 0.781   | 0.054   | 0.219   |
>
>
> We should emphasize however that this is a minor point that can easily be modified if this is a concern. For example, would the practitioner be interested in the delay and area obtainable under the optimal recipe, one could imagine during label generation to run logic synthesis with 10 different recipes, and keeping the smallest obtained delay and area as labels. Our model trained on such labels would then produce an estimate of the best area and delay obtainable for a given circuit. Or, if one wanted flexibility in prediction, one could imagine adding the parameters that control the synthesis recipe as arguments of the model, so that the model would produce estimates of area and delay conditional on specific choices of synthesis parameters. We did not consider such use-cases in our work to not overburden the text, but they would be easily accommodated if the need arose.

---

> > ### Comment · Reviewer_fovb · 2024-11-28
> >
> > Thank you for your response and the updated manuscript. I have decided to increase my review score from 5 to 6.

---

### Official Review · Reviewer_ZT1D · 2024-10-30

**Soundness:** 3
**Presentation:** 3
**Contribution:** 3
**Rating:** 6
**Confidence:** 2

**Summary:**

Introduction of a new ML framework: VeraDistill to predict circuit QoR metrics like area/delay bypassing costly logic synthesis processes.  The framework uses a verilog-trained LLM (codeV-7B) to produce a representation (final layer) to feed into the FFN to make predictions for the QoR metrics during inference.  To train, they take the AIG LUT outputs of the synthesis flow and use GNNs to produce embeddings where the GNN acts as a "teacher" resulting in knowledge distillation. The loss metric used for final training is a simple MSE involving not only the FFN predictions and synthesis QoR data, but also the final layer data from the GCN(teacher) and LLM (Student).  The LLM is frozen in the training process.

**Strengths:**

As a hardware engineer, I greatly appreciate the contributions in the work as this can provide a lot of benefit to practioners.

I find the knowledge distillation approach between the GNNs and LLMs to be  novel. The work showcases strengths of GNNs and LLMs in a creative fashion. The work also shows that LLMs have remarkably learned about the underlying circuit representation underneath the verilog code.

The results are also impressive, showing improvements over baselines including the out-of-distribution datasets like OpenABCD.

**Weaknesses:**

Its unclear to me how reliant the framework is on a particular LLM (CodeV) and how versatile a single instance (trained) will be in accommodating more unseen RTL code. The authors do acknowledge the lack of Verilog/RTL code for training, so while not a weakness per se, it does make me question the generalization capabilities.

Not much details are shared about the training resources (outside of the 8 V100 GPUs used).

**Questions:**

1) Does the knowledge distillation help with a specific subset of Verilog designs (more than others)?  Could be interesting to exploit this.
2) Please elaborate on the training times
3)  There are many ways to code (e.g an adder) in Verilog, does the framework handle this well?
4) Curious to hear whether you think this approach can scale to handle other RTL (e.g VHDL)

---

> ### Author Response · Authors · 2024-11-26
>
> Thank you very much for your review, and we particularly appreciate hearing a practitioner agree with the real-life potential of such a model! Here are answers to your comments.
>
> Alternative LLMs: To demonstrate performance of VeriDistill with different LLMs, we employ CodeV-DeepSeek and CodeV-CodeQwen, which utilize deepseek-coder-6.7b and CodeQwen1.5-7B-Chat as the base models. These two models are two variants of the standard CodeV model based on CodeLlama-7b-Instruct, and were trained with the same procedure. As can be seen from the table, the final performance is only marginally different from using different Verilog LLMs. We have included this result in the appendix.
>
> The performance of VeriDistill with different Large Language Models:
>
> | Approach                      | MAE ↓   | R2 ↑    | MAPE ↓  | RSE ↓   | MAE ↓   | R2 ↑    | MAPE ↓  | RSE ↓   |
> |-------------------------------|---------|---------|---------|---------|---------|---------|---------|---------|
> | CodeQwen + Decoder             | 1.070   | 0.563   | 1.975   | 0.437   | 0.766   | 0.322   | 0.150   | 0.678   |
> | DeepSeek + Decoder             | 1.061   | 0.566   | 2.184   | 0.434   | 0.759   | 0.340   | 0.149   | 0.660   |
> | CodeV + Decoder                | 0.991   | 0.614   | 1.901   | 0.386   | 0.718   | 0.443   | 0.141   | 0.557   |
> | VeriDistill (CodeQwen)         | 0.468   | 0.878   | 0.574   | 0.122   | 0.424   | 0.733   | 0.078   | 0.267   |
> | VeriDistill (DeepSeek)         | 0.484   | 0.875   | 0.622   | 0.125   | 0.426   | 0.706   | 0.077   | 0.294   |
> | VeriDistill (CodeV)            | 0.495   | 0.862   | 0.629   | 0.138   | 0.415   | 0.728   | 0.076   | 0.272   |
>
>
> Training resources: Because the LLM is kept frozen during training, it was possible to save training time by extracting the forward pass through the LLM only once and saving it. We performed this phase on a machine with 8 Nvidia V100 GPUs with 32GB of memory and 32 Intel(R) Xeon(R) Gold 6140 CPUs. Once the hidden state was saved, we then trained each model following the procedure detailed in the paper on the same machine using a single V100 GPU with 1024 minibatch sizes. The training times for each model are summarized in the following table.
>
> Training times for the various models:
> | Method          | Training Time (Till Convergence) | Number of Epochs to Converge |
> |-----------------|----------------------------------|-----------------------------|
> | LUT-GNN         | 21 hours                         | 300                         |
> | AST-XGBoost     | 5 minutes                        | N/A                         |
> | AST-GNN         | 33 minutes                       | 340                         |
> | AST-GNN w/ KD   | 40 minutes                       | 300                         |
> | CodeV + Decoder | 12 hours                         | 360                         |
> | VeriDistill     | 18 hours                         | 260                         |
>
>
> We have added this description and the table in the appendix.
>
> Question 1: This is a very interesting question, and we had attempted to investigate this by looking into the Verilog code where VeriDistill made the largest errors. In general, we could not find particular patterns in the code for the errors. We noticed however that the model tended to do worse on instances with large memories, e.g. one example was the declaration of a memory with 4096 16-bit elements "reg [15:0]mem[0:4095]" in a SRAM design. It is possible that Verilog LLM was not well trained to comprehend the impact of large memory size on circuit delay, leading to underpredict the delay. If this is true, then addition of more such cases in pre-training data would likely enhance the performance of VeriDistill on these cases.
>
> Question 2: Please see answer to "Training resources" above.
>
> Question 3: Our training set consists of Verilog instances of various complexity. The simplest instances contain various implementations of common building blocks in Verilog (such as adders, subtractors, different logical gates, flip-flops, counters, etc.). Given that VeriDistill achieves a high accuracy on these simpler instances (as shown in figure 4), we can attest that VeriDistill can easily handle basic functionalities specified in Verilog with near-zero error.
>
> Question 4: Yes, absolutely, we do not see any obstacles in principle. However, the recent HDL LLMs are mostly trained with Verilog data parsed from Github and textbooks. While VHDL is more popular in Europe, there is not as much VHDL data available in the web, which limits the development of LLMs for VHDL designs. If the VHDL data is not an issue, possibly for some universities or large corporations, we believe that the proposed approach could handle VHDL just as well.

---

> ### Author Response · Authors · 2024-12-02
>
> Thank you for your detailed review and the insightful questions you posed. Could you please provide additional feedback if there are any aspects we have not fully addressed?
> Thank you for your time and effort in helping us enhance our work.

---

### Official Review · Reviewer_DJeE · 2024-11-04

**Soundness:** 2
**Presentation:** 2
**Contribution:** 2
**Rating:** 3
**Confidence:** 4

**Summary:**

The main contributions of this work, as reported, are:

A. Development of a VeriDistill framework: The first truly end-to-end machine learning model that processes raw Verilog code directly, without preprocessing, to accurately estimate circuit area and delay metrics.

B. Innovative Knowledge Distillation (KD): A novel knowledge distillation method is applied during training, transferring low-level circuit insights (as LUT graphs) back into the model for enhanced predictions. Experiments show that this approach surpasses previous state-of-the-art (SOTA) baselines on a large-scale Verilog dataset, with improved generalization to out-of-distribution (OOD) data.  The use of both LLM representations and knowledge distillation is critical to the model’s performance, as omitting either component decreases performance below baseline levels.

**Strengths:**

The overall research motivation is well-articulated, and the proposed idea is both interesting and has promising applications.

**Weaknesses:**

1. The manuscript lacks clear description of the proposed approach. Both figure 1 and figure 2 should be improved in both schematic and description. Methodology section 3.1 must be elaborated. I will advice to break the training strategies and steps at first and finally combine them towards final goal/results. It will enhance readability, understanding of the general audience with more detailed explanation of the intermediate training flows. Figure 1 & 2 and section 3.3 description seems disjointed. Reproducibility from this text description is problematic.

2. in Section 4.1 EXPERIMENTAL SETUP,   "The model’s decoder feedforward network depicted
in Fig. ?? is designed so that the 512-dimensional representations from the...", Fig. reference is missing.

3. The overall quality of writing and presentation could be improved. The writing should maintain a smoother flow, as some paragraphs feel disjointed. For instance, in the experimental setup section 4.1 , certain details—such as the explanation of 512-dimensional state tokens—are important for understanding the methodology and would benefit from clearer integration.

4.  It is recommended to compare the performance of the proposed model with previous models/approaches to better emphasize its novelty, particularly in terms of computational complexity, data preprocessing demands, resource utilization, and the metrics of area and delay. For instance, could you provide an analytical comparison of your approach against the work by Fang et al. (2023/2024b), addressing the computational time for SOG and the complexity of converting linguistic data into bit-level operators using the logic synthesis tool Yosys (Wolf et al., 2013)? This comparison would further clarify the advantages of your approach, especially given that accuracy, precision, and sensitivity are crucial metrics for circuit quality estimation in addition to computational efficiency.

5. In Fig. 4, please clarify the presence of scattered outlier points for both area (lower values) and delay (higher values) in relation to the fitting curve. These sparse data points appear to indicate cases where VeriDistill’s performance aligns with that of other baseline models. Could you explain why VeriDistill performs similarly to the baselines for these specific outliers, and discuss any factors that might contribute to this behavior in terms of model limitations or specific data characteristics?

6. Authors explained "Finally, in Figure 5, we present the t-SNE projection of the last hidden space representations on the test data from the teacher model (Z_teacher) trained for predicting log-area, alongside those from the LLM-based models. As can be seen, the resulting t-SNE representation of the VerDistill model appears very similar to the one of the LUT-GNN teacher model, albeit slightly rotated by roughly 30 degrees to the right (which is arbitrary and immaterial). The AST-GNN with KD model also can be seen as a rotation of the LUT-GNN plot, roughly 90 degrees to the left. In contrast, the plot of the CodeV + Decoder appears much more like an undefined mass."

It seems hypothetical assumption about the statement "VerDistill model appears very similar to the one of the LUT-GNN teacher model, albeit slightly rotated by roughly 30 degrees to the right (which is arbitrary and immaterial). The AST-GNN with KD model also can be seen as a rotation of the LUT-GNN plot, roughly 90 degrees to the left.".  AST-GNN with KD is quite different than LUT-GNN and VeriDistill, as well it does not seem "t-SNE representation of the VerDistill model appears very similar to the one of the LUT-GNN teacher model"...and if the authors knew already "slightly rotated by roughly 30 degrees to the right".. please adjust this rotation to make it fit overlapped and compared.
""
7. Please be consistent with proposed approach name .......authors mentioned "VerDistill" in one place and "VeriDistill" in other place(s). Please correct any typos. Also, grammatical errors have been noticed.

**Questions:**

This has already been addressed in the Weaknesses section.

---

> ### Author Response · Authors · 2024-11-26
>
> We thank the reviewer for the thorough comments.
>
> Weakness 1:
> Following your comments, we rewrote and reorganized the methodology section. The model is now clearly differentiated from the training strategy, which is introduced in two steps. We have also redone Figures 1 and 2, to better match the text description. We hope this addresses the issues.
>
> Weaknesses 2--3:
> We rewrote Section 4.1 accordingly.
>
> Weakness 4:
> We were not able to run MasterRTL (Fang et al. 2023/2024b) as a baseline because its feature extraction step relies on a path-level slack prediction model, which is not provided in the opensourced repository. We did nonetheless try to compare to the SOG representation proposed in their work, which is an intermediate representation between RTL and netlist. However, the SOG extraction process, which relies on Yosys, PyVerilog and a Verilog-to-graph parser, failed to produce SOGs for a large proportion (27\%) of our dataset, comprised almost entirely of the designs with large delays in the dataset. This emphasizes the complexity of relying on third-party tools for pre-processing Verilog files. On the other hand, our proposed VeriDistill takes the design Verilog files as inputs without any pre-processing requirement, and is able to provide design quality estimation instantaneously during inference. Moreover, the rich representation obtained from LLMs can be fused effectively with teacher GNN embedding, which is not the case for the GNN-based representation as shown in the AST-KD setting.
>
>
> Weakness 5:
> We had attempted to investigate the outliers by looking into the Verilog code where VeriDistill made the largest errors. In general, we could not find particular patterns in the code for the errors. We noticed however that the model tended to do worse on instances with large memories, e.g. one example was the declaration of a memory with 4096 16-bit elements "reg [15:0]mem[0:4095]" in a SRAM design. It is possible that Verilog LLM was not well trained to comprehend the impact of large memory size on circuit delay, leading to underpredict the delay. If this is true, then addition of more such cases in pre-training data would likely enhance the performance of VeriDistill on these cases.
>
> In addition, we would like to point out that even though there are sparse outlier data points in all plots in Fig 4, VeriDistill has much fewer outlier points and performs generally better on these outlier data points. To illustrate this, we outline the distribution of absolute errors across all methods in  (full results can be found in Appendix 5). While the baselines AST-XGBoost, AST-GNN, AST-GNN w/ KD, and CodeV + Decoder have 26, 33, 30, 47 points with absolute error larger than 0.5, respectively, VeriDistill only has 17 such points.
>
> Distribution of absolute percentage errors for the (log) delay prediction task. Each cell specifies the number of points with the absolute percentage error falling in the range specified by the column:
> | Method          | 0 - 0.1 | 0.1 - 0.3 | 0.3 - 0.5 | 0.5 - 1.0 |
> |-----------------|---------|-----------|-----------|-----------|
> | AST-XGBoost     | 1572    | 695       | 81        | 26        |
> | AST-GNN         | 1402    | 799       | 141       | 33        |
> | AST-GNN w/ KD   | 1298    | 891       | 153       | 30        |
> | CodeV + Decoder | 1127    | 1002      | 192       | 47        |
> | VeriDistill     | 1803    | 504       | 52        | 17        |
>
>
> Weakness 6:
> We thank the reviewer for pointing out the error in our explanation for the t-SNE plot. We have made changes to the text to address the errors and clarify our explanation.
>
> Typographical and grammatical errors:
> We made a new pass and corrected everything.

---

> ### Author Response · Authors · 2024-12-02
>
> We wanted to follow up on our previous response and the improvements we've made based on your insightful review. As the rebuttal period nears its end, we would like to ensure that we have addressed all your concerns. If so, we would be grateful if you could reconsider your rating of our paper.
>
> Thank you for your time and effort in helping us enhance our work.
>
> We look forward to your response.

---

### Author Response · Authors · 2024-11-26

We thank the reviewers for their thorough insights. In particular, we are pleased that the reviewers both appreciated the fully end-to-end nature of our model (from raw Verilog to QoR metrics!) and the practical potential value of such a model.

We hope to have addressed the questions and concerns raised in our individual responses. We have also updated the manuscript with the modifications highlighted in red for clarity.

---

### Meta-Review · Area_Chair_GjbZ · 2024-12-21

**Metareview:**

**Summary:** The paper introduces VeriDistill, a framework for predicting circuit quality metrics from raw Verilog code, bypassing traditional logic synthesis. The approach employs a knowledge distillation framework where a graph neural network (GNN) teacher trained on Look-Up Table (LUT) graphs guides a Verilog-trained large language model (LLM) student. The model demonstrates competitive performance against baseline methods, including AST-GNNs and XGBoost.


**Strength:**
1. This paper makes a significant contribution as the first to implement knowledge transfer for circuit quality estimation from the netlist level to the RTL level.

2. The distillation framework, which transfers GNNs’ predictions on the netlist to LLMs that process raw Verilog code, is both intuitive and effective.

3. The proposed method demonstrates strong performance across benchmarks, including generalization to out-of-distribution datasets.

**Weakness:**
1. The major concern is the lack of justification for using LLM-based circuit quality estimators over GNN-based ones. Based on Table 1 of this submission, the former demonstrates lower accuracy than the latter and may face efficiency challenges when processing long Verilog designs. The authors did not provide quantitative results to justify the practical applicability of LLM-based methods.

2. The proposed method is evaluated only on CodeV, lacking demonstrations on more diverse LLMs and scalability to large-scale circuit designs.

3. Clarity and writing issues, including missing figures and insufficient details about the methodology and training resources, negatively impact reproducibility.


**Reasons for the decision:**

While VeriDistill addresses an important problem and demonstrates improved experimental results, its lack of analysis regarding advantages over GNN-based methods, questionable scalability, and clarity issues diminish its potential impact. Therefore, I am inclined to recommend rejection.

**Additional Comments On Reviewer Discussion:**

During the rebuttal period, the key concerns raised by the reviewers and the corresponding author response are summarized as follows:

**1. Scalability to large designs**

Reviewer Concern (zReM, ZT1D): The framework was criticized for its limited scalability due to LLM token constraints, making it less applicable to large industrial designs.

Author Response: The authors argued that advancements in LLMs with longer context lengths could alleviate this issue in future implementations. However, no quantitative results are provided regarding this aspect.

**2. Methodology clarity and writing issues**

Reviewer Concern (DJeE, fovb): Missing figure references, unclear descriptions of the methodology, and general disjointedness in writing were highlighted as barriers to understanding and reproducibility.

Author Response: The authors rewrote sections of the manuscript, improved the figures, and corrected typographical errors.

**3. Teacher model performance vs. student**

Reviewer Concern (zReM, ZT1D, fovb): The performance gap between the GNN teacher and the LLM student was raised as a concern, questioning the practical utility of the approach, particularly given the inefficiency of the latter when processing long-context inputs.

Author Response: The authors emphasized the value of bypassing costly synthesis in their method.

The authors’ rebuttal improved the manuscript’s clarity and addressed some concerns about scalability. However, significant limitations remain, particularly in the comparison with GNN-based approaches regarding both accuracy and scalability. These unresolved issues have influenced the decision to recommend rejection.

---

### Decision · Program_Chairs · 2025-01-22

Reject